# Intra and inter-rater reproducibility of the Remote Static Posture Assessment (ARPE) protocol's Postural Checklist

**Betiane Moreira Pilling**[1]*, **Cláudia Tarragô Candotti**[1], **Marcelle Guimarães Silva**[1], **Marina Ziegler Frantz**[1], **Matias Noll**[2]

**1** Programa de Pós-Graduação de Ciências do Movimento Humano, UFRGS, Porto Alegre, RS, Brazil, **2** Child and Adolescent Health Research Group (GPSaCA), Instituto Federal Goiano, Goiás, Brazil3 Universidade Federal de Goiás, Goiás, Brazil

* betianefisioterapeuta@gmail.com

**Data Availability Statement:** All relevant data are within the paper and its Supporting Information files.

## Abstract

With the enforcement of social distancing due to the pandemic, a need to conduct postural assessments through remote care arose. So, this study aimed to assess the intra- and inter-rater reproducibility of the Remote Static Posture Assessment (ARPE) protocol's Postural Checklist. The study involved 51 participants, with the postural assessment conducted by two researchers. For intra-rater reproducibility assessment, one rater administered the ARPE protocol twice, with an interval of 7–days between assessments (test–retest). A second independent rater assessed inter-rater reproducibility. Kappa statistics ($k$) and percentage agreement (%C) were used, with a significance level of 0.05. The intra-rater reproducibility analysis indicated high reliability, $k$ values varied from 0.921 to 1.0, with %C ranging from 94% to 100% for all items on the ARPE protocol's Postural Checklist. Inter-rater reproducibility indicates reliability ranging from slight to good, $k$ values exceeded 0.4 for the entire checklist, except for four items: waists in the frontal photograph ($k = 0.353$), scapulae in the rear photograph ($k = 0.310$), popliteal line of the knees in the rear photograph ($k = 0.270$), and foot posture in the rear photograph ($k = 0.271$). Nonetheless, %C surpassed 50% for all but the scapulae item (%C = 47%). The ARPE protocol's Postural Checklist is reproducible and can be administered by the same or different raters for static posture assessment. However, when used by distinct raters, the items waists (front of the frontal plane), scapulae, popliteal line of the knees, and feet (rear of the frontal plane) should not be considered.

## Introduction

During the coronavirus disease 2019 pandemic, a need arose to develop methods for delivering patient care remotely [1,2]. The World Health Organization defines Telehealth as "the use of information and communication technologies (ICTs) to deliver health services at a distance" [3,4]. In this context, virtual or remote care is a practical solution to offer essential services during social distancing and reduce non-essential clinical services [1,5,6].

**Funding:** This publication has financial support for publication from the Federal Institute Goiano-Brazil. The funders had no role in study design, data collection and analysis, decision to publish, or preparation of the manuscript.

**Competing interests:** The authors have declared that no competing interests exist.

Developing clinical protocols and therapeutic guidelines for remote physiotherapy care has become indispensable. In this context, research in telerehabilitation (remote care) aids in enhancing evidence-based practices, refining distance assessment procedures, and optimizing the rehabilitation process [1,3,7]. Thus, a primary challenge in implementing remote-based physiotherapy is ensuring the ability to conduct valid and reliable assessments, such as postural assessments. Validity and reproducibility are essential measurement properties that must underpin selecting and developing tools for clinical, educational, or research practices [8]. Various methods are used to evaluate static body posture, from the observation of anatomical points, to the use of photographs and their measurements [9]. The results of these assessments guide the prescription of exercises, monitoring of progress and treatment [10]. A recent scoping review revealed studies showcasing various tools for assessing body posture for remote assessment [11]. However, only three of these studies provided the measurement properties of these tools [11].

The Remote Static Posture Assessment (ARPE) protocol has been recently developed and validated to address this gap. This protocol involves acquiring photographs through remote means that are then interpreted using a Postural Checklist [12]. Concerning the reliability of these remote postural assessments, the ARPE protocol still needs research to demonstrate the precision of the results obtained by different raters (inter-rater reproducibility) and by assessments conducted by the same rater at different times (intra-rater reproducibility) [8,13]. Given that the results of the postural assessment are crucial in determining the decisions that form the basis of the therapeutic plan for postural deviations, it is essential to understand these measurement properties before integrating this protocol into remote physiotherapy practice [8,14–16].

Considering the importance of evaluating the reliability of these measures, this study aims to assess the intra- and inter-rater reproducibility of the Remote Static Posture Assessment (ARPE) protocol's Postural Checklist. Moreover, it is hypothesized that the ARPE protocol can be used by the same evaluator or different evaluators to remotely evaluate static posture.

## Materials and methods

The reproducibility of the ARPE protocol was assessed following two Guidelines: (1) COSMIN – *Consensus-Based Standards for the Selection of Health Measurement Instruments* [17], which guides researchers in conducting studies to assess measurement properties; and (2) GRRAS–- Guideline for Reporting Reliability and Agreement Studies [18]. This study received approval from the Ethics and Research Committee of the University where it was conducted (CAAE: 54077321.1.0000.5347).

Two researchers, both physiotherapists with professional experience in postural assessment ranging from 5 to 15 years, served as raters (A and B). These raters, blind to each other's assessments, underwent specific training using the ARPE protocol. This training spanned over 20 hours and encompassed seven pilot assessments, which were not included in the main sample. The training of the evaluators initially consisted of discussing (1) each item of the Postural Checklist and (2) the pilot assessments. This procedure had been aim of ensuring uniformity of all steps when using the Postural Checklist.

The non-probabilistic and consecutive sample comprised 51 participants recruited via social media and the "snowball" methodology, which consists of a non-probabilistic sampling technique in which existing study subjects they recruit future subjects from among their acquaintances [19]. The inclusion criteria for participants were age 18 years or older, ability to stand unassisted by orthoses during the assessment, and no prostheses in their lower limbs. These criteria were established with the aim of reducing aspects that could interfere with the

results of the assessment in relation to body balance. The exclusion criterion was a failure to participate in any assessment sessions. Recruitment began on 05/19/2022 and ended on 09/29/2022, and participants completed a written informed consent form.

Participants who reached out to the researchers following the disclosure of the research were sent the following via *WhatsApp*: (a) a video, lasting 3 minutes [20], explaining the research; (b) the Participant's Manual, a PDF file containing basic instructions for preparing for the remote postural assessment; and (c) a link to the Informed Consent Form (ICF). Once the ICF was completed, the date and time for the remote assessment were scheduled.

An hour before the assessment day, participants were sent a *WhatsApp* message containing a link to a *Zoom* meeting for the remote assessment. Only Rater A and the research participant attended this *Zoom* meeting. Rater A was in charge of capturing the assessment video. This video features the participant standing upright and is recorded from both frontal (front and rear views) and sagittal planes, with each view lasting approximately 10 seconds.

To acquire the video, the participant must be dressed in skimpy clothing, a bikini for women and swim trunks for men, with hair tied up and feet barefoot. The person being evaluated was instructed to position themselves in front, back and side of the camera, which should be positioned at half the height of the person being evaluated. All information for carrying out the assessment is contained in the subject and assessor manual and in the explanatory video that are part of the ARPE protocol [12]. During the study, the confidentiality of personal data was ensured, with the collected data being anonymized. These data were stored to maintain the confidentiality and privacy of the research participants' information. A USB drive was used to share the images.

### Intra-rater reproducibility

The intra-rater reproducibility of the ARPE's Postural Checklist was assessed by the same rater (Rater A), who collected and analyzed the data at two different times, with an interval ranging from 7 to 10 days between assessments. This interval was determined based on previous research [21–24], ensuring that the duration was neither too long nor too short, thus minimizing the potential for rater interpretation bias. At each assessment session, Rater A sequentially: (a) captured the desired image (either from the rear, front, or side) from the assessment video; (b) superimposed a virtual plumb line on the captured image (photograph); and (c) analyzed the posture of the participants in the captured photographs using the *Postural Checklist*. These procedures were repeated for images captured from the rear, front, and side.

### Inter-rater reproducibility

Inter-rater reproducibility was assessed by having one Rater B analyze a video. The results from Rater B's assessment were then compared to the results from the initial assessment by Rater A. Both assessments were performed independently of each other (Fig 1). Rater B received the video recorded during the remote consultation from Rater A via a USB drive. After obtaining the videos, Rater B captured images from the video, positioned the virtual plumb line, and analyzed the participants' posture in the photographs using the *Postural Checklist*. These procedures were repeated for images captured from the rear, front, and side. The raters were instructed to assess the entire sample on a single plane before proceeding to another assessment plane.

### Statistical analysis

The sample size of 51 participants for the ARPE Checklist to be considered reproducible was determined using G-Power 3.1, based on the methods of Sim and Wright [21], using a two-

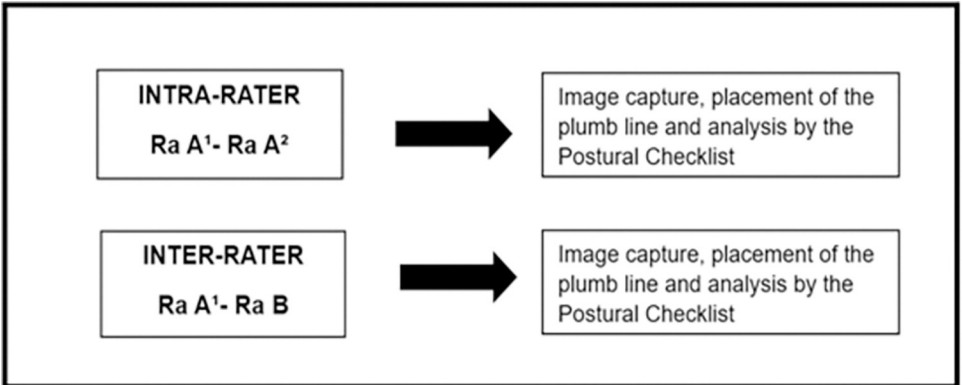

**Fig 1. Reproducibility of the research design (Rater A$^1$ = first assessment by rater A; Rater A$^2$ = second assessment by rater A, conducted 7 to 10 days later).**

tailed test that assumed a null hypothesis of $k = 0.4$ and an alternative hypothesis of $k = 0.8$, with a significance level of $\alpha = 0.05$, power of 80% and loss of 10%.

Data were analyzed using the Statistical Package for the Social Sciences (SPSS), version 20.0, and Excel, version 2016. The variables were described using absolute and relative frequencies, mean and standard deviation to characterize the sample.

To ensure reproducibility, the Kappa statistics ($k$) and the percentage of agreement (%C) were used [18]. The classification of $k$ values was based on Altman [25], where $k$ values $< 0.20$ are considered as "poor"; 0.21–0.40 as "mild"; 0.41–0.60 as "moderate"; 0.61–0.80 as "good"; and 0.81–1.00 as "very good". The classification of %C values was based on Janse *et al.* [26], where values of %C $< 0.30$ are considered as "poor"; 0.31–0.50 as "weak"; 0.51–0.70 as "moderate"; 0.71–0.90 as "good"; and 0.91–1.00 as "excellent". A $k$ value $> 0.40$ (indicating moderate concordance) and a %C value $> 0.50$ (indicating moderate concordance) were set as the criteria to determine whether the ARPE protocol checklist is reproducible. The significance level was set at $p < 0.05$.

## Results

Fifty-one participants were involved in this study (Table 1).

Additional information on sample characterization is contained in S1 Appendix.

The intra-rater reproducibility analysis indicated high reliability. The $k$-value ranged from 0.921 to 1.0. Additionally, the percentage of agreement (%C) was excellent, ranging from 94% to 100%, and all values were statistically significant (Table 2).

Inter-rater reproducibility indicates reliability ranging from mild to good, with $k$ values ranging from 0.270 to 0.795. The %C values range from poor to good, between 47% and 88%.

**Table 1. Sample characterization (n = 51).**

| Characteristic | Description |
|---|---|
| Age | 29 ± 10 years |
| Gender | 73% Female (n = 37) |
| | 27% Male (n = 14) |
| Body Mass | 66.7 ±14.1 Kg |
| Height | 166.0 ± 9.3 cm |
| BMI | 24.2 ± 4.0 Kg/m$^2$ |

**Table 2. Results of the intra-rater reproducibility analysis of the ARPE's Postural Checklist (Rater A assessed it twice, with an interval of 7 days between assessments).**

| Postural Variable from the Checklist | Kappa (k) | Confidence Interval (CI 95%) | Percentage Agreement (%C) | p-value |
|---|---|---|---|---|
| 1. FF Global Examination | 1 | 1.000_1.000 | 100 | <0.001 |
| 2. FF Head Tilt | 0.967 | 0.901_1.033 | 98 | <0.001 |
| 3. FF Head Rotation | 1 | 1.000_1.000 | 100 | <0.001 |
| 4. FF Shoulders | 1 | 1.000_1.000 | 100 | <0.001 |
| 5. FF Waists | 0.969 | 0.907_1.031 | 98 | <0.001 |
| 6. FF Knees | 1 | 1.000_1.000 | 100 | <0.001 |
| 7. RF Global Examination | 1 | 1.000_1.000 | 100 | <0.001 |
| 8. RF Scapulae | 0.921 | 0.833_1.009 | 94 | <0.001 |
| 9. RF Waists | 0.970 | 0.912_1.028 | 98 | <0.001 |
| 10. RF Popliteal Lin. of the Knees | 1 | 1.000_1.000 | 100 | <0.001 |
| 11. RF Knees | 0.971 | 0.913_1.029 | 98 | <0.001 |
| 12. RF Feet | 1 | 1.000_1.000 | 100 | <0.001 |
| 13. SP Global Examination | 1 | 1.000_1.000 | 100 | <0.001 |
| 14. SP Head | 1 | 1.000_1.000 | 100 | <0.001 |
| 15. SP Dorsal Col. | 1 | 1.000_1.000 | 100 | <0.001 |
| 16. SP Dorsal Col. | 1 | 1.000_1.000 | 100 | <0.001 |
| 17. SP Pelvic Tilt | 0.928 | 0.830_1.026 | 96 | <0.001 |
| 18. SP Pelvic Thrust | 1 | 1.000_1.000 | 100 | <0.001 |
| 19. SP Knees | 1 | 1.000_1.000 | 100 | <0.001 |

Rater A assessed it at two different times, with an interval of 7 to 10 days between assessments

FF: front of the frontal plane; RF: rear of the frontal plane; SP: sagittal plane; Lin.: line; Col.: column.

It is noteworthy that the *k* values were below 0.4 for only four postural variables, and the %C was below 50% only for the variable "RF Scapulae" (scapulae assessed in the rear of the frontal plane) (Table 3). These four items should not be assessed by the ARPE protocol when different raters perform the assessment.

## Discussion

To the best of our knowledge, this study is original, and the results demonstrate that the checklist can be used remotely. It is, therefore, a suitable instrument for teleservice, with high intra-rater reliability. Furthermore, it has from slight to good inter-evaluator reliability, except for four items that should not be used: waistline in the frontal photograph, scapulae in the rear photograph, popliteal line of the knees in the back photograph and foot posture in the back photograph.

Many professionals who observe anatomical points for qualitative postural assessment often neglect methodological standards that guarantee the consistent replication of findings, hindering future comparisons [27]. Based on this assumption, the ARPE Postural Checklist presents itself as a method that allows, through the conceptual description of the terms alignment, misalignment and alteration, that the evaluator can be guided to classify the posture of the person evaluated. Although the Posture Checklist does not provide quantitative details such as angular or linear values of body proportions, it does allow for a comprehensive assessment of static posture. It also provides guidelines and reference points to assist the evaluator in interpreting postural observations, facilitating the comparison and identification of asymmetries in the frontal and sagittal planes, even without any marking on the evaluator, which is

**Table 3. Results of the inter-rater reproducibility analysis of the ARPE's Postural Checklist.**

| Postural Variable from the Postural Checklist | Rater A x Rater B | | | |
|---|---|---|---|---|
| | Kappa (*k*) | Confidence Interval (CI 95%) | Percentage Agreement (%C) | p-value |
| 1. FF Global Examination | 0.785 | 0.633_0.937 | 86 | <0.001 |
| 2. FF Head Tilt | 0.604 | 0.418_0.790 | 76 | <0.001 |
| 3. FF Head Rotation | 0.532 | 0.342_0.722 | 73 | <0.001 |
| 4. FF Shoulders | 0.699 | 0.529_0.869 | 80 | <0.001 |
| 5. FF Waists | 0.353 | 0.167_0.539 | 55 | <0.001 |
| 6. FF Knees | 0.795 | 0.653_0.937 | 86 | <0.001 |
| 7. RF Global Examination | 0.784 | 0.634_0.934 | 86 | <0.001 |
| 8. RF Scapulae | 0.310 | 0.152_0.468 | 47 | <0.001 |
| 9. RF Waists | 0.586 | 0.410_0.762 | 73 | <0.001 |
| 10. RF Popliteal Lin. of the Knees | 0.270 | 0.048_0.492 | 67 | <0.001 |
| 11. RF Knees | 0.793 | 0.651_0.935 | 86 | <0.001 |
| 12. RF Feet | 0.271 | 0.071_0.471 | 55 | <0.001 |
| 13. SP Global Examination | 0.759 | 0.573_0.945 | 88 | <0.001 |
| 14. SP Head | 0.548 | 0.356–0.740 | 78 | <0.001 |
| 15. SP Dorsal Col. | 0.653 | 0.463_0.843 | 80 | <0.001 |
| 16. SP Lumbar Col. | 0.422 | 0.212_0.632 | 65 | <0.001 |
| 17. SP Pelvic Tilt | 0.749 | 0.581_0.917 | 86 | <0.001 |
| 18. SP Pelvic Thrust | 0.677 | 0.455_0.899 | 86 | <0.001 |
| 19. SP Knees | 0.524 | 0.304_0.744 | 75 | <0.001 |

; FF: front of the frontal plane; RF: rear of the frontal plane; SP: sagittal plane; Lin: line; Col.: column.

essential for remote care. The observation of anatomical points is a commonly used method, easily applicable to any ergonomic or clinical study where the qualitative assessment of static posture is the main objective. This technique allows the mapping of postural deviations in various degrees and levels of severity [28,29].

In this context, assessing measurement properties is crucial for evidence-based practice in physiotherapy [30–32]. Understanding the measurement properties of assessment methods aids in decision-making regarding diagnosis, prognosis, treatment planning, and setting expected results. Moreover, it facilitates comprehension of the obtained results [32].

The reproducibility assessment of the ARPE protocol's Postural Checklist was based on the notion that the reliability of remote assessments is determined by the accuracy of results obtained either by different raters or from assessments performed at different times [8]. According to the results, the ARPE protocol can be used for intra- and inter-rater assessments, provided that instructions from both the participant's and rater's manuals are followed. These guidelines cover aspects ranging from setting up the environment and preparing the participant, to record and capture the image, to the use of the Postural Checklist and other vital information.

In assessing intra-rater reproducibility (the same rater at different times), values of *k* and %C above 0.92 and 94%, respectively, indicate an "almost perfect agreement" [21], which confirms that the ARPE protocol's Postural Chec*k*list is reproducible for all items. The ARPE is a convenient, practical, and effective postural assessment protocol, suitable for situations where face-to-face contact is not feasible. The advantages of ARPE are that it does not need an assistant alongside the participant, and it does not require any markings on the participant's body. However, even with excellent intra-rater results, it is important to note that postural assessment relies heavily on the rater's experience [33,34].

For assessing inter-rater reproducibility (with different raters assessing the same participant), prior training exceeding 20 hours was undertaken, along with seven pilot assessments. The choice for this training approach was based on recognizing the importance of adequately training raters for reproducibility studies [18]. The raters, who operated independently and were blinded to each other's assessments, were instructed to use the Postural Checklist by plane. In other words, they were tasked with assessing the entire sample on one plane before proceeding to the next. This procedure was chosen to minimize assessment bias related to memory. According to Sim and Wright [21], memory bias and rater blinding can influence the obtained results. Nonetheless, despite the rigorous methodology employed in this study, four of the nineteen postural variables exhibited $k$ values below 0.4: "FF Waists ($k = 0.353$)", "Popliteal line of the knees ($k = 0.270$)", "Feet ($k = 0.271$)", and "RF Scapulae ($k = 0.310$)". Only the variable "RF Scapulae" (%C 47%) had a %C less than 50%. Although the ARPE can be used by different evaluators, the items Waist in the frontal plane, Scapulas, Popliteal line of the knees and Feet should not be evaluated. Further research is needed to assess the inter-rater reproducibility of these items. This recommendation has been added to the ARPE protocol header in S2 Appendix.

Regarding the inter-rater assessment of the Scapulae, where both $k$ and %C yielded results below what is considered reproducible, we believe that the clothing worn by the women played a significant role. In the Participant's Manual, they were asked not to wear a swimmer-style top to expose the spine adequately. However, when a straight-cut top was worn, it obscured the inferior angle of the scapulae, hindering clear visualization for analysis. Several authors [33–36] highlight the importance of appropriate clothing to ensure clear visibility of the structures under assessment. Therefore, for future studies, it would be ideal for the assessed women to wear bikinis with a thin strap at the back to enhance visibility, especially of the scapulae. For our results, when evaluating the Waist, Popliteal line of the knees and Feet, we suspect that the image quality (photography resolution) may have been responsible for the lower-than-expected values.

Therefore, one of the challenges in assessing posture through remote assistance is to improve the inferior technical quality of the photographs obtained. This poor image quality could stem from the camera's capabilities, internet connectivity issues, or the patient's difficulty setting up a suitable assessment environment, including choosing an appropriate location and positioning the camera and themselves [8,33,37]. Therefore, this factor should be considered when selecting ARPE as it presents a distinct scenario from in-person posture assessments conducted with photographs. In the latter, the images captured are of high quality, devoid of distortions (parallax and zoom), and possess both clarity and the necessary scale to allow for detailed observation and contrast. Consequently, this ensures that the individual's body details are clearly visible in the photograph, preventing the rater from making errors in their assessment [22,38,39].

Regarding certain technical concerns, providing a video tutorial before the postural assessment has been recommended to assist the patient [35,37]. This tutorial is intended to prepare the patient for the evaluation. To reduce complications during the assessment with ARPE, a 2-minute and 40-second video [20] was sent to the study participants to reinforce the guidelines provided in the Participant's Manual. In a recent scoping review, aimed at identifying tools that assess static posture via telecare, none of the three articles offered a manual to guide either the participants or the raters [11].

The ARPE protocol is a tool that evaluates static posture. Nonetheless, it is recognized that postural assessment is broad and complex to rely solely on the *Postural Checklist*. Therefore, in clinical practice, it is advised to supplement the assessment performed with ARPE with tests for flexibility, muscle strength, and balance. This advice stems from the understanding that various factors influence body posture [24,40,41].

## Limitations

The main challenges encountered during this study are related to the technical issues associated with remote care, including slow internet connections leading to frozen images and voice disconnections, challenges in positioning the camera, additional noise, power outages, and battery depletion in the participant's device. The quality of the images obtained has a great impact on the use of the ARPE protocol. Therefore, both the evaluator and the person being evaluated must follow all the guidelines contained in the manuals and informative video to obtain the video, with the aim of ensuring the capture of the best possible image. Nevertheless, remote assessments are essential despite these challenges, given the growing prevalence of remote care in clinical settings.

## Conclusions

The ARPE protocol's Postural Checklist has demonstrated both intra- and inter-rater reproducibility. This means it can be used by the same rater or by different raters. Based on the $k$ and %C values in intra-evaluator reproducibility data, where the same professional who evaluates your patient before, during and after treatment, intra-evaluator reliability guarantees the usability of the ARPE. Regarding inter-evaluator reproducibility, we recommend caution when evaluating the following items of the ARPE assessment: waist in front of the frontal plane, scapulae, popliteal line of the knees and feet behind the frontal plane. However, the exclusion of 4 items is only in case the instrument needs to be used in a context where different evaluators will analyze the patient's posture, such as in a postural assessment service. Still, only 4 items should be viewed with caution. This recommendation has been added to the ARPE protocol header in S2 Appendix.

Despite the ARPE protocol including a Participant's Manual, a Rater's Manual, an explanatory video, and a Postural Checklist, its use for assessing static posture via telecare needs further research in terms of testing concurrent validity.

## Supporting information

**S1 Appendix. Additional information on sample characterization.**
(TIF)

**S2 Appendix. Header of the Postural Checklist.**
(TIF)

## Author Contributions

**Conceptualization:** Betiane Moreira Pilling, Cláudia Tarragô Candotti, Marcelle Guimarães Silva.

**Data curation:** Betiane Moreira Pilling, Marcelle Guimarães Silva, Marina Ziegler Frantz.

**Formal analysis:** Betiane Moreira Pilling, Cláudia Tarragô Candotti.

**Methodology:** Betiane Moreira Pilling, Cláudia Tarragô Candotti, Matias Noll.

**Project administration:** Betiane Moreira Pilling, Cláudia Tarragô Candotti.

**Supervision:** Betiane Moreira Pilling, Cláudia Tarragô Candotti.

**Validation:** Betiane Moreira Pilling, Cláudia Tarragô Candotti.

**Writing – original draft:** Betiane Moreira Pilling, Cláudia Tarragô Candotti.

**Writing – review & editing:** Betiane Moreira Pilling, Cláudia Tarragô Candotti, Marcelle Guimarães Silva, Marina Ziegler Frantz, Matias Noll.

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
