## [Decision Letter · Decision Letter 0]

8 Nov 2023

PONE-D-23-28429Intra and Inter-rater Reproducibility of the Remote Static Posture Assessment (ARPE) Protocol's Postural ChecklistPLOS ONE

Dear Dr. Pilling,

Thank you for submitting your manuscript to PLOS ONE. After careful consideration, we feel that it has merit but does not fully meet PLOS ONE’s publication criteria as it currently stands. Therefore, we invite you to submit a revised version of the manuscript that addresses the points raised during the review process.

We look forward to receiving your revised manuscript.

Kind regards,

Ravi Shankar Yerragonda Reddy, Ph.D

Academic Editor

PLOS ONE

3. Please amend your authorship list in your manuscript file to include authors Betiane Moreira Pilling, Cláudia Tarragô Candotti, Marcelle Guimarães Silva, Marina Ziegler Frantz, and Matias Noll.

Additional Editor Comments:

Title: Reproducibility of the Remote Static Posture Assessment (ARPE)

Abstract:

The abstract provides a clear overview of the study's background, objectives, methods, and key findings. However, it lacks specific details about the results, making it difficult for readers to grasp the significance of the study at a glance.

Introduction:

1. The introduction provides a comprehensive background on the importance of remote physiotherapy care during the COVID-19 pandemic. However, it could benefit from more concise language to maintain the reader's attention and focus on the research question.

2. While the introduction sets the stage well, it could be more specific about the existing gaps in the literature related to remote postural assessments and why the ARPE protocol is needed.

3. The transition from the general context to the specific research objectives could be smoother to clarify the study's purpose more effectively.

Methods:

1. The methods section is detailed and well-structured, providing a clear understanding of the study design and procedures. However, there are some areas that could be improved:

- The rationale for choosing specific inclusion criteria and the significance of these criteria should be explained in more detail.

- The training of the raters is briefly mentioned but could be expanded to describe the specific aspects of the ARPE protocol they were trained on and how their training ensured consistency.

- The method of recruitment via social media and the "snowball" methodology could benefit from a brief explanation for readers who may not be familiar with these terms.

- The section on data analysis mentions using SPSS and Excel, but it lacks information on the specific statistical tests used, which should be included for transparency.

Results:

1. The results section presents the key findings of the study, and while it provides important information about the reproducibility of the ARPE protocol, there are several points that require attention:

- The sample size and characteristics are adequately described in Table 1, which is helpful for readers to understand the study's population.

- The presentation of intra-rater reproducibility results in Table 2 is clear and detailed, showing high reliability (k values above 0.92) and excellent percentage of agreement (%C) values for all items.

- However, the presentation of inter-rater reproducibility results in Table 3 is less clear. While it mentions that four postural variables exhibited k values below 0.4 and %C values below 50%, it doesn't provide a clear explanation or discussion of the implications of these findings. The reasons for these lower reproducibility values need to be addressed.

- The discussion of technical challenges related to remote assessments is insightful, but it would be helpful to elaborate on how these challenges may have influenced the results, especially the lower reproducibility values for certain variables.

- The limitations section briefly mentions technical challenges but could benefit from further discussion regarding their potential impact on the results. Additionally, it might be valuable to discuss any potential limitations related to the study's sample characteristics, such as age and gender distribution, and their relevance to the reproducibility findings.

2. The presentation of results is generally clear, but the discussion of lower reproducibility values in inter-rater assessments lacks depth and explanation. It's essential to provide insights into why certain variables had lower reproducibility and how these findings may affect the practical use of the ARPE protocol in clinical settings.

3. The conclusion summarizes the key findings but could be strengthened by discussing the practical implications of the results in more detail. Specifically, it should address how the recommendations to exclude certain items from the ARPE assessment impact its usability in telecare and remote clinical practice.

Reviewers' comments:

Reviewer's Responses to Questions

**Comments to the Author**

1. Is the manuscript technically sound, and do the data support the conclusions?

Reviewer #1: Yes

Reviewer #2: Partly

2. Has the statistical analysis been performed appropriately and rigorously? 

Reviewer #1: Yes

Reviewer #2: N/A

3. Have the authors made all data underlying the findings in their manuscript fully available?

Reviewer #1: Yes

Reviewer #2: No

4. Is the manuscript presented in an intelligible fashion and written in standard English?

Reviewer #1: Yes

Reviewer #2: Yes

5. Review Comments to the Author

Reviewer #1: This is a study that addresses an interesting issue: telerehabilitation-based assessments. In general, the manuscript is well written, although the authors should review the coherence linking some paragraphs, specially in introduction and discussion sections.

Some minor comments are appendend below:

- Keywords: This reviewer encourages authors to add more keywords (as long as the platform allows it), such as balance, postural assesment, etc.

INTRODUCTION SECTION:

- Developing clinical protocols and therapeutic guidelines for remote physiotherapy care has become indispensable. This deepens discussions regarding the criteria associated with this clinical practice. In this context, research in telerehabilitation (remote care) aids in enhancing evidence-based practices, refining distance assessment procedures, and optimizing the rehabilitation process [1,2,6].: This paragraph is a bit repetitive and confusing, especially the sentence "This deepens discussions regarding the criteria associated with this clinical practice.". It gives the feeling of a "filler paragraph".

Thus, a primary challenge in implementing remote-based physiotherapy is ensuring the ability to conduct valid and reliable assessments, such as postural assessments. Validity and reproducibility are essential measurement properties that must underpin selecting and developing tools for clinical, educational, or research practices [7].: It is too short a paragraph. This reviewer suggests linking it somehow with the previous paragraph because there is no continuity in the reading.

- The Remote Static Posture Assessment (ARPE) protocol has been recently developed and validated to address this gap. This protocol involves acquiring photographs through remote means that are then interpreted using a Postural Checklist [8]. Concerning the reliability of these remote postural assessments, the ARPE protocol still needs research to demonstrate the precision of the results obtained by different raters (inter-rater reproducibility) and by assessments conducted by the same rater at different times (intra-rater reproducibility) [7,9]. Given that the results of the postural assessment are crucial in determining the decisions that form the basis of the therapeutic plan for postural deviations, it is essential to understand these measurement properties before integrating this protocol into remote physiotherapy practice [7,10-12].: This reviewer recommends the authors to develop other aspects such as balance, posture, etc.

- This study aims to assess the intra- and inter-rater reproducibility of the Remote Static Posture Assessment (ARPE) protocol's Postural Checklist.: Please link the objective of the study to the previous paragraph and present a hypothesis..

METHODS SECTION:

- Sample size: Please move the sample size information from the beginning of the material and methods section to the statistical analysis section and inform about the participants that you need for having statistically significant differences.

- Inclusion and exclusion criteria: the authors indicate that they included adults over 18 years of age, but do not specify whether an age limit was established. It should be taken into account that the older the age, the less stability and therefore the more confounding factors. On the other hand, the absence of prosthesis should be considered as an exclusion criterion, i.e., people with a prosthesis were excluded. It is strange to see that there was only one exclusion criterion.

- Please add the subsection "statistical analysis". Also, significance should be estimated through p-value.

RESULTS SECTION:

- Fifty-one participants were involved in this study (Table 1).: Please, elaborate and describe the sample. You cannot simply give a sentence in this regard. Altough it is detailed in table 1, you must describe at least some things of your sample in a paragraph. Also, you need a flow chart of the participants.

DISCUSSION SECTION:

- To the best of our knowledge, the study is original, and the results demonstrate that the checklist can be used remotely, making it suitable for teleassistance.: Please elaborate a little more on this paragraph. The first paragraph of the discussion should report the results and whether or not the hypotheses and objectives of the study have been met.

- While the Postural Checklist does not offer quantitative details, such as angular or linear values of body proportions, it enables a comprehensive assessment of static posture. It also gives directions and reference points to help the rater interpret postural observations, making it easier to compare and identify asymmetries in both the frontal and sagittal planes, even without any markings on the person being assessed, which is crucial for remote care. Observing anatomical points is a commonly used method, easily applicable to any ergonomic or clinical study where the qualitative assessment of static posture is the primary objective. This technique enables the mapping of postural deviations across varying degrees and levels of severity [21,22]. However, many professionals who observe anatomical points for qualitative postural assessment often neglect methodological standards that ensure the consistent replication of findings, impairing future comparisons [23].: This paragraph seems to be taken from the literature rather than a conclusion of the authors.

- A search of the PubMed, Scopus, and Embase databases revealed 794 studies showcasing various tools for assessing body posture suited for remote assessment. However, only three of these studies provided the measurement properties of these tools [27].: This is not a systematic review. This information is superfluous.

Reviewer #2: Dear authors,

Abstract

-Generally acceptable.

Introduction

-Generally acceptable. Can you explain in a paragraph why we are doing posture analysis in this section? You can also give brief information about other methods used for posture analysis.

Materials and Methods

- Can you give some more details about how the measurements were made? How were they dressed? From which directions were measurements taken (lateral, frontal or back side) How was the camera positioned? Measurements were taken from a distance of several meters. These need to be standardized.

Results

Discussion:

- You need to discuss your results a little more in the Discussion section. You need to comment further on the 4 areas with low measurement validity.

6. PLOS authors have the option to publish the peer review history of their article (what does this mean?). If published, this will include your full peer review and any attached files.

Reviewer #1: No

Reviewer #2: **Yes: **Esedullah AKARAS

---

## [Author Response · Author response to Decision Letter 0]

22 Dec 2023

Editorial comments:

Answer: We apologize for the oversight. We reviewed the entire text, inserting the appropriate nomenclature.

2. We note that the grant information you provided in the ‘Funding Information’ and ‘Financial Disclosure’ sections do not match

Answer: We apologize for the mistake and included funding from the Federal Institute of Goiano.

3. Please amend your authorship list in your manuscript file to include authors Betiane Moreira Pilling, Cláudia Tarragô Candotti, Marcelle Guimarães Silva, Marina Ziegler Frantz, and Matias Noll.

Answer: We apologize for the mistake and have inserted the names of the authors and their affiliations after the title of the manuscript.

Answer: We have inserted supporting information at the end of the manuscript and renamed “appendix 1” to “appendix A1”.

Additional Editor Comments:

Title: Reproducibility of the Remote Static Posture Assessment (ARPE)

Abstract:

The abstract provides a clear overview of the study's background, objectives, methods, and key findings. However, it lacks specific details about the results, making it difficult for readers to grasp the significance of the study at a glance.

Answer: Thanks for your important suggestion. We inserted specific details regarding about the results in the abstract in order to facilitate readers’ understanding.

Introduction:

1. The introduction provides a comprehensive background on the importance of remote physiotherapy care during the COVID-19 pandemic. However, it could benefit from more concise language to maintain the reader's attention and focus on the research question.

Answer: We agree with the reviewer. Based on your suggestions, we have reformulated the introduction with more concise language.

2. While the introduction sets the stage well, it could be more specific about the existing gaps in the literature related to remote postural assessments and why the ARPE protocol is needed.

Answer: As requested, we have inserted into the introduction the existing gaps in the literature related to remote postural assessments and why the ARPE protocol is necessary.

Page 3,4, lines 64-66: 

“A recent scoping review revealed 794 studies showcasing various tools for assessing body posture for remote assessment [11]. However, only three of these studies provided the measurement properties of these tools [11]. “ 

3. The transition from the general context to the specific research objectives could be smoother to clarify the study's purpose more effectively.

Answer: We've reworked the transition from general context to the present objectives.

Page 4 , lines 78-82: 

“Considering the importance of evaluating the reliability of these measures, this study aims to assess the intra- and inter-rater reproducibility of the Remote Static Posture Assessment (ARPE) protocol's Postural Checklist. Moreover, it is hypothesized that the ARPE protocol can be used by the same evaluator or different evaluators to remotely evaluate static posture.”

Methods:

1. The methods section is detailed and well-structured, providing a clear understanding of the study design and procedures. 

Answer: Thanks for your positive feedback.

However, there are some areas that could be improved:

- The rationale for choosing specific inclusion criteria and the significance of these criteria should be explained in more detail.

Answer: Thanks for your in-depth review. The rationale for the specific inclusion criteria was included in the Material and Methods, as follow:.

Page 5, lines 102-109: 

“The inclusion criteria for participants were age 18 years or older, ability to stand unassisted by orthoses during the assessment, and no prostheses in their lower limbs. These criteria were established with the aim of reducing aspects that could interfere with the results of the assessment in relation to body balance. The exclusion criterion was a failure to participate in any assessment sessions. Recruitment began on 05/19/2022 and ended on 09/29/2022, and participants completed a written informed consent form.”.

- The training of the raters is briefly mentioned but could be expanded to describe the specific aspects of the ARPE protocol they were trained on and how their training ensured consistency.

Answer: Training for evaluators was included in the Materials and Methods to ensure consistency, as follow:

Page 5, lines 91-98: 

“Two researchers, both physiotherapists with professional experience in postural assessment ranging from 5 to 15 years, served as raters (A and B). These raters, blind to each other's assessments, underwent specific training using the ARPE protocol. This training spanned over 20 hours and encompassed seven pilot assessments, which were not included in the main sample. The training of the evaluators initially consisted of discussing (1) each item of the Postural Checklist and (2) the pilot assessments. This procedure had be aim of ensuring uniformity all steps when using the Postural Checklist. ”

- The method of recruitment via social media and the "snowball" methodology could benefit from a brief explanation for readers who may not be familiar with these terms.

Answer: An explanation of the "snowball" methodology was included in the Materials and Methods, as follow:

Page 5, lines 99-102: 

“The non-probabilistic and consecutive sample comprised 51 participants recruited via social media and the "snowball" methodology, which consists of a non-probabilistic sampling technique in which existing study subjects they recruit future subjects from among their acquaintances [19].”

- The section on data analysis mentions using SPSS and Excel, but it lacks information on the specific statistical tests used, which should be included for transparency.

Answer: Sorry for our mistake. We inserted a description about data analysis in the Statistical Analysis, as follow:

Page 8, lines 165-170:

“Data were analyzed using the Statistical Package for the Social Sciences (SPSS), version 20.0, and Excel, version 2016. The variables were described using absolute and relative frequencies, mean and standard deviation to characterize the sample. To ensure reproducibility, the Kappa statistics (k) and the percentage of agreement (%C) were used [18]. 

Results:

1. The results section presents the key findings of the study, and while it provides important information about the reproducibility of the ARPE protocol, there are several points that require attention:

- The sample size and characteristics are adequately described in Table 1, which is helpful for readers to understand the study's population.

Answer: Thanks for your positive feedback.

- The presentation of intra-rater reproducibility results in Table 2 is clear and detailed, showing high reliability (k values above 0.92) and excellent percentage of agreement (%C) values for all items.

Answer: Thanks for your positive feedback.

- However, the presentation of inter-rater reproducibility results in Table 3 is less clear. While it mentions that four postural variables exhibited k values below 0.4 and %C values below 50%, it doesn't provide a clear explanation or discussion of the implications of these findings. The reasons for these lower reproducibility values need to be addressed.

Answer: We have included in the Discussion the reasons for these lower reproducibility values and the implications of these findings, as follow:

Page 15, lines 270-287: 

“Although the ARPE can be used by different evaluators, the items Waist in the frontal plane, Scapulas, Popliteal line of the knees and Feet should not be evaluated. Further research is needed to assess the inter-rater reproducibility of these items. This recommendation has been added to the ARPE protocol header in Appendix A2.

Regarding the inter-rater assessment of the Scapulae, where both k and %C yielded results below what is considered reproducible, we believe that the clothing worn by the women played a significant role. In the Participant's Manual, they were asked not to wear a swimmer-style top to expose the spine adequately. However, when a straight-cut top was worn, it obscured the inferior angle of the scapulae, hindering clear visualization for analysis. Several authors [33-36] highlight the importance of appropriate clothing to ensure clear visibility of the structures under assessment. Therefore, for future studies, it would be ideal for the assessed women to wear bikinis with a thin strap at the back to enhance visibility, especially of the scapulae. For our results, when evaluating the Waist, Popliteal line of the knees and Feet, we suspect that the image quality (photography resolution) may have been responsible for the lower-than-expected values.”

- The discussion of technical challenges related to remote assessments is insightful, but it would be helpful to elaborate on how these challenges may have influenced the results, especially the lower reproducibility values for certain variables.

Answer: It was included in the Discussion how technical challenges may have influenced the results of the variables with lower values, Scapulas, Waist, Poplites Line of the knees and Feet.

Page 15, lines 270-284: 

“Although the ARPE can be used by different evaluators, the items Waist in the frontal plane, Scapulas, Popliteal line of the knees and Feet should not be evaluated. Further research is needed to assess the inter-rater reproducibility of these items.

Regarding the inter-rater assessment of the Scapulae, where both k and %C yielded results below what is considered reproducible, we believe that the clothing worn by the women played a significant role. In the Participant's Manual, they were asked not to wear a swimmer-style top to expose the spine adequately. However, when a straight-cut top was worn, it obscured the inferior angle of the scapulae, hindering clear visualization for analysis. Several authors [33-36] highlight the importance of appropriate clothing to ensure clear visibility of the structures under assessment. Therefore, for future studies, it would be ideal for the assessed women to wear bikinis with a thin strap at the back to enhance visibility, especially of the scapulae. 

Page 17, lines 320-323: 

“The quality of the images obtained has a great impact on the use of the ARPE protocol. Therefore, both the evaluator and the person being evaluated must follow all the guidelines contained in the manuals and informative video to obtain the video, with the aim of ensuring the capture of the best possible image.”

- The limitations section briefly mentions technical challenges but could benefit from further discussion regarding their potential impact on the results. Additionally, it might be valuable to discuss any potential limitations related to the study's sample characteristics, such as age and gender distribution, and their relevance to the reproducibility findings.

Answer: We understand the reviewer's point of view and improve the description of the limitations of our study.The impact of technical difficulties was included in the Discussion, as follow:

Page 17, lines 320-323: 

“The quality of the images obtained has a great impact on the use of the ARPE protocol. Therefore, both the evaluator and the person being evaluated must follow all the guidelines contained in the manuals and informative video to obtain the video, with the aim of ensuring the capture of the best possible image.”

2. The presentation of results is generally clear, but the discussion of lower reproducibility values in inter-rater assessments lacks depth and explanation. It's essential to provide insights into why certain variables had lower reproducibility and how these findings may affect the practical use of the ARPE protocol in clinical settings.

Answer: We have included in the Discussion the reasons for these lower reproducibility values and the implications of these findings, as follow:

Page 15, lines 270-287: 

“Although the ARPE can be used by different evaluators, the items Waist in the frontal plane, Scapulas, Popliteal line of the knees and Feet should not be evaluated. Further research is needed to assess the inter-rater reproducibility of these items.

Regarding the inter-rater assessment of the Scapulae, where both k and %C yielded results below what is considered reproducible, we believe that the clothing worn by the women played a significant role. In the Participant's Manual, they were asked not to wear a swimmer-style top to expose the spine adequately. However, when a straight-cut top was worn, it obscured the inferior angle of the scapulae, hindering clear visualization for analysis. Several authors [33-36] highlight the importance of appropriate clothing to ensure clear visibility of the structures under assessment. Therefore, for future studies, it would be ideal for the assessed women to wear bikinis with a thin strap at the back to enhance visibility, especially of the scapulae. For our results, when evaluating the Waist, Popliteal line of the knees and Feet, we suspect that the image quality (photography resolution) may have been responsible for the lower-than-expected values.”

Page 17, lines 320-323: 

“The quality of the images obtained has a great impact on the use of the ARPE protocol. Therefore, both the evaluator and the person being evaluated must follow all the guidelines contained in the manuals and informative video to obtain the video, with the aim of ensuring the capture of the best possible image.”

3. The conclusion summarizes the key findings but could be strengthened by discussing the practical implications of the results in more detail. Specifically, it should address how the recommendations to exclude certain items from the ARPE assessment impact its usability in telecare and remote clinical practice.

Answer: The impact of the usability of the ARPE protocol on remote clinical practice was included in the Conclusions , as follow:

Page 19, lines 327-339: 

“Based on the k and %C values in intra-evaluator reproducibility data, where the same professional who evaluates your patient before, during and after treatment, intra-evaluator reliability guarantees the usability of the ARPE. Regarding inter-evaluator reproducibility, we recommend caution when evaluating the following items of the ARPE assessment: waist in front of the frontal plane, scapulae, popliteal line of the knees and feet behind the frontal plane. However, the exclusion of 4 items is only in case the instrument needs to be used in a context where different evaluators will analyze the patient's posture, such as in a postural assessment service. Still, only 4 items should be viewed with caution. This recommendation has been added to the ARPE protocol header in Appendix A2.”.

Reviewer #1: 

This is a study that addresses an interesting issue: telerehabilitation-based assessments. In general, the manuscript is well written, although the authors should review the coherence linking some paragraphs, specially in introduction and discussion sections.

Answer: Thanks for your positive feedback and for your efforts to help us to improve this manuscript. The introduction and discussion sections were revised based on all reviewers comments.

Some minor comments are appendend below:

- Keywords: This reviewer encourages authors to add more keywords (as long as the platform allows it), such as balance, postural assesment, etc.

Answer: We appreciated the suggestion and the Keywords “postural balance” and the term “postural assessment” were included in the keywords, as well as the other terms: Epidemiology. Men’s Health. Population Health. Public health. Spine. Women’s Health.

INTRODUCTION SECTION:

- Developing clinical protocols and therapeutic guidelines for remote physiotherapy care has become indispensable. This deepens discussions regarding the criteria associated with this clinical practice. In this context, research in telerehabilitation (remote care) aids in enhancing evidence-based practices, refining distance assessment procedures, and optimizing the rehabilitation process [1,2,6].: This paragraph is a bit repetitive and confusing, especially the sentence "This deepens discussions regarding the criteria associated with this clinical practice.". It gives the feeling of a "filler paragraph".

Thus, a primary challenge in implementing remote-based physiotherapy is ensuring the ability to conduct valid and reliable assessments, such as postural assessments. Validity and reproducibility are essential measurement properties that must underpin selecting and developing tools for clinical, educational, or research practices [7].: It is too short a paragraph. This reviewer suggests linking it somehow with the previous paragraph because there is no continuity in the reading.

Answer: Thanks for your in-depth review. We rewrote and improved this paragraph, as follow:

Page 3, lines 52-60: 

“Developing clinical protocols and therapeutic guidelines for remote physiotherapy care has become indispensable. In this context, research in telerehabilitation (remote care) aids in enhancing evidence-based practices, refining distance assessment procedures, and optimizing the rehabilitation process [1,3,7]. Thus, a primary challenge in implementing remote-based physiotherapy is ensuring the ability to conduct valid and reliable assessments, such as postural assessments. Validity and reproducibility are essential measurement properties that must underpin selecting and developing tools for clinical, educational, or research practices [8] ”.

- The Remote Static Posture Assessment (ARPE) protocol has been recently developed and validated to address this gap. This protocol involves acquiring photographs through remote means that are then interpreted using a Postural Checklist [8]. Concerning the reliability of these remote postural assessments, the ARPE protocol still needs research to demonstrate the precision of the results obtained by different raters (inter-rater reproducibility) and by assessments conducted by the same rater at different times (intra-rater reproducibility) [7,9]. Given that the results of the postural assessment are crucial in determining the decisions that form the basis of the therapeutic plan for postural deviations, it is essential to understand these measurement properties before integrating this protocol into remote physiotherapy practice [7,10-12].: This reviewer recommends the authors to develop other aspects such as balance, posture, etc.

Answer: We understand and agree with the reviewer's point of view, as clinical decision-making to draw up a postural treatment plan must be based on several aspects, such as: patient history and complaint, assessment of postural balance, assessment of static and dynamic misalignments of the body in the space, assessment of strength, flexibility and body self-perception. We have highlighted the importance of evaluating the reproducibility measurement properties of the ARPE protocol to evaluate "static posture" through remote care. Many other studies are needed to improve postural assessment as a whole and to address all relevant aspects to outline patient conduct, monitoring or results related to body posture.

- This study aims to assess the intra- and inter-rater reproducibility of the Remote Static Posture Assessment (ARPE) protocol's Postural Checklist.: Please link the objective of the study to the previous paragraph and present a hypothesis..

Answer: Thanks for your in-depth review. We rewrote and improved this paragraph, as follow:

Page 4, lines 78-82: 

“Considering the importance of evaluating the reliability of these measures, this study aims to assess the intra- and inter-rater reproducibility of the Remote Static Posture Assessment (ARPE) protocol's Postural Checklist. And it is hypothesized that the ARPE protocol can be used by the same evaluator or different evaluators to remotely evaluate static posture.”.

METHODS SECTION:

- Sample size: Please move the sample size information from the beginning of the material and methods section to the statistical analysis section and inform about the participants that you need for having statistically significant differences.

Answer: Thanks for your in-depth review. We moved and rewrote this paragraph, as follow:

Page 7-8, lines 160-164: 

“The sample size of 51 participants for the ARPE Checklist to be considered reproducible was determined using G-Power 3.1, based on the methods of Sim and Wright [16], using a two-tailed test that assumed a null hypothesis of k = 0.4 and an alternative hypothesis of k = 0.8, with a significance level of α=0.05, power of 80% and loss of 10%.”.

- Inclusion and exclusion criteria: the authors indicate that they included adults over 18 years of age, but do not specify whether an age limit was established. It should be taken into account that the older the age, the less stability and therefore the more confounding factors. On the other hand, the absence of prosthesis should be considered as an exclusion criterion, i.e., people with a prosthesis were excluded. It is strange to see that there was only one exclusion criterion.

Answer: A maximum age was not established because one of the requirements for inclusion in the study was that the subject had the ability to stand, thus excluding individuals with little stability. The absence of users of orthoses and prosthetics as an inclusion factor, as they were not included in the sample. An explanation of the exclusion criteria was included.

- Please add the subsection "statistical analysis". Also, significance should be estimated through p-value.

Answer: We rewrote base on your suggestion, as follow:

Page 7-8, lines 159-178 

“Statistical Analysis

The sample size of 51 participants for the ARPE Checklist to be considered reproducible was determined using G-Power 3.1, based on the methods of Sim and Wright [16], using a two-tailed test that assumed a null hypothesis of k = 0.4 and an alternative hypothesis of k = 0.8, with a significance level of α=0.05, power of 80% and loss of 10%.

Data were analyzed using the Statistical Package for the Social Sciences (SPSS), version 20.0, and Excel, version 2016. The variables were described using absolute and relative frequencies, mean and standard deviation to characterize the sample. To ensure reproducibility, the Kappa statistics (k) and the percentage of agreement (%C) were used [14]. The classification of k values was based on Altman [19], where k values < 0.20 are considered as "poor"; 0.21–0.40 as "mild"; 0.41–0.60 as "moderate"; 0.61–0.80 as "good"; and 0.81–1.00 as "very good". The classification of %C values were based on JANSE et al. [20], where values of %C < 0.30 are considered as "poor"; 0.31–0.50 as "weak"; 0.51–0.70 as "moderate"; 0.71–0.90 as "good"; and 0.91–1.00 as "excellent". A k value > 0.40 (indicating moderate concordance) and a %C value > 0.50 (indicating moderate concordance) were set as the criteria to determine whether the ARPE protocol checklist is reproducible. The significance level was set at p < 0.05’. 

RESULTS SECTION:

- Fifty-one participants were involved in this study (Table 1).: Please, elaborate and describe the sample. You cannot simply give a sentence in this regard. Altough it is detailed in table 1, you must describe at least some things of your sample in a paragraph. Also, you need a flow chart of the participants.

Answer: We agree with the reviewer that this information was missing. As requested, we have included more information and a flowchart in the Appendix A1.

DISCUSSION SECTION:

- To the best of our knowledge, the study is original, and the results demonstrate that the checklist can be used remotely, making it suitable for teleassistance.: Please elaborate a little more on this paragraph. The first paragraph of the discussion should report the results and whether or not the hypotheses and objectives of the study have been met.

Answer: Thanks for your suggestion. Based on it, we have inserted the following sentences:

Page 12, lines 211-217: 

“To the best of our knowledge, this study is original, and the results demonstrate that the Postural Checklist can be used remotely. It is, therefore, a suitable instrument for teleservice, with high intra-rater reliability. Furthermore, it has from slight to good inter-evaluator reliability, except for four items that should not be used: waistline in the frontal photograph, scapulae in the rear photograph, popliteal line of the knees in the back photograph and foot posture in the back photograph”.

- While the Postural Checklist does not offer quantitative details, such as angular or linear values of body proportions, it enables a comprehensive assessment of static posture. It also gives directions and reference points to help the rater interpret postural observations, making it easier to compare and identify asymmetries in both the frontal and sagittal planes, even without any markings on the person being assessed, which is crucial for remote care. Observing anatomical points is a commonly used method, easily applicable to any ergonomic or clinical study where the qualitative assessment of static posture is the primary objective. This technique enables the mapping of postural deviations across varying degrees and levels of severity [21,22]. However, many professionals who observe anatomical points for qualitative postural assessment often neglect methodological standards that ensure the consistent replication of findings, impairing future comparisons [23].: This paragraph seems to be taken from the literature rather than a conclusion of the authors.

Answer: Thanks for your important consideration. The paragraph was readjusted and supplemented, as follow:

Page 12,13, lines 218-233: 

“Many professionals who observe anatomical points for qualitative postural assessment often neglect methodological standards that guarantee the consistent replication of findings, hindering future comparisons [27]. Based on this assumption, the ARPE Postural Checklist presents itself as a method that allows, through the conceptual description of the terms alignment, misalignment and alteration, that the evaluator can be guided to classify the posture of the person evaluated. Although the Posture Checklist does not provide quantitative details such as angular or linear values of body proportions, it does allow for a comprehensive assessment of static posture. It also provides guidelines and reference points to assist the evaluator in interpreting postural observations, facilitating the comparison and identification of asymmetries in the frontal and sagittal planes, even without any marking on the evaluator, which is essential for remote care. The observation of anatomical points is a commonly used method, easily applicable to any ergonomic or clinical study where the qualitative assessment of static posture is the main objective. This technique allows the mapping of postural deviations in various degrees and levels of severity [28-29].

- A search of the PubMed, Scopus, and Embase databases revealed 794 studies showcasing various tools for assessing body posture suited for remote assessment. However, only three of these studies provided the measurement properties of these tools [27].: This is not a systematic review. This information is superfluous.

Answer: Thanks for your important consideration. The information was inserted in the introduction, as follow:

Page 3-4, lines 64-66: “A recent scoping review revealed 794 studies showcasing various tools for assessing body posture for remote assessment. However, only three of these studies provided the measurement properties of these tools [27]. 

Reviewer #2: 

Dear authors,

Abstract

-Generally acceptable.

Answer: Thanks for your positive feedback. Based on the other reviewer we made some improvements.

Introduction

-Generally acceptable. Can you explain in a paragraph why we are doing posture analysis in this section? You can also give brief information about other methods used for posture analysis.

Answer: Thanks for your important suggestion. Based on it, a paragraph has been inserted in the Introduction about why to analyze posture and information about other methods used for posture analysis, as follow:

Page 3, lines 60-64: 

“ Various methods are used to evaluate static body posture, from the observation of anatomical points, to the use of photographs and their measurements [9].And the results of these assessments guide the prescription of exercises, monitoring of progress and treatment [10].

Materials and Methods

- Can you give some more details about how the measurements were made? How were they dressed? From which directions were measurements taken (lateral, frontal or back side) How was the camera positioned? Measurements were taken from a distance of several meters. These need to be standardized.

Answer: We understand that our text was not clear enough and we included a brief description of the standardization of image capture in the Materials and methods (page 7, lines (153-155). In detail, all the details regarding how to carry out the evaluation using the ARPE protocol are contained in the evaluated and evaluator manuals and in the explanatory video.

Discussion:

- You need to discuss your results a little more in the Discussion section. You need to comment further on the 4 areas with low measurement validity.

Answer: The information was inserted in the discussion, as follow:

Page 15, lines 270-287: 

“Although the ARPE can be used by different evaluators, the items Waist in the frontal plane, Scapulas, Popliteal line of the knees and Feet should not be evaluated. Further research is needed to assess the inter-rater reproducibility of these items.Regarding the inter-rater assessment of the Scapulae, where both k and %C yielded results below what is considered reproducible, we believe that the clothing worn by the women played a significant role. In the Participant's Manual, they were asked not to wear a swimmer-style top to expose the spine adequately. However, when a straight-cut top was worn, it obscured the inferior angle of the scapulae, hindering clear visualization for analysis. Several authors [28,29,33,34] highlight the importance of appropriate clothing to ensure clear visibility of the structures under assessment. Therefore, for future studies, it would be ideal for the assessed women to wear bikinis with a thin strap at the back to enhance visibility, especially of the scapulae. For our results, when evaluating the Waist, Popliteal line of the knees and Feet, we suspect that the image quality (photography resolution) may have been responsible for the lower-than-expected values.”

---

## [Decision Letter · Decision Letter 1]

8 Jan 2024

Intra and Inter-rater Reproducibility of the Remote Static Posture Assessment (ARPE) Protocol's Postural Checklist

PONE-D-23-28429R1

Dear Dr. Betiane Moreira Pilling,

We’re pleased to inform you that your manuscript has been judged scientifically suitable for publication and will be formally accepted for publication once it meets all outstanding technical requirements.

Kind regards,

Ravi Shankar Yerragonda Reddy, Ph.D

Academic Editor

PLOS ONE

Reviewers' comments:

Reviewer's Responses to Questions

**Comments to the Author**

1. If the authors have adequately addressed your comments raised in a previous round of review and you feel that this manuscript is now acceptable for publication, you may indicate that here to bypass the “Comments to the Author” section, enter your conflict of interest statement in the “Confidential to Editor” section, and submit your "Accept" recommendation.

Reviewer #2: All comments have been addressed

2. Is the manuscript technically sound, and do the data support the conclusions?

Reviewer #2: Yes

3. Has the statistical analysis been performed appropriately and rigorously? 

Reviewer #2: Yes

4. Have the authors made all data underlying the findings in their manuscript fully available?

Reviewer #2: Yes

5. Is the manuscript presented in an intelligible fashion and written in standard English?

Reviewer #2: Yes

6. Review Comments to the Author

Reviewer #2: All corrections were made by the authors.

7. PLOS authors have the option to publish the peer review history of their article (what does this mean?). If published, this will include your full peer review and any attached files.

Reviewer #2: **Yes: **Esedullah AKARAS

---

## [Editor Report · Acceptance letter]

1 Feb 2024

PONE-D-23-28429R1 

PLOS ONE

Dear Dr. Pilling, 

I'm pleased to inform you that your manuscript has been deemed suitable for publication in PLOS ONE. Congratulations! Your manuscript is now being handed over to our production team.

Kind regards, 

on behalf of

Dr. Ravi Shankar Yerragonda Reddy 

Academic Editor

PLOS ONE